# Metabolism of Selenium, Selenocysteine, and Selenoproteins in Ferroptosis in Solid Tumor Cancers

**DOI:** 10.3390/biom12111581

**Published:** 2022-10-28

**Authors:** Briana K. Shimada, Sydonie Swanson, Pamela Toh, Lucia A. Seale

**Affiliations:** Pacific Biosciences Research Center, University of Hawaii at Manoa, Honolulu, HI 96822, USA

**Keywords:** selenium, ferroptosis, cancer, solid tumors, selenium metabolism

## Abstract

A potential target of precision nutrition in cancer therapeutics is the micronutrient selenium (Se). Se is metabolized and incorporated as the amino acid selenocysteine (Sec) into 25 human selenoproteins, including glutathione peroxidases (GPXs) and thioredoxin reductases (TXNRDs), among others. Both the processes of Se and Sec metabolism for the production of selenoproteins and the action of selenoproteins are utilized by cancer cells from solid tumors as a protective mechanism against oxidative damage and to resist ferroptosis, an iron-dependent cell death mechanism. Protection against ferroptosis in cancer cells requires sustained production of the selenoprotein GPX4, which involves increasing the uptake of Se, potentially activating Se metabolic pathways such as the trans-selenation pathway and the TXNRD1-dependent decomposition of inorganic selenocompounds to sustain GPX4 synthesis. Additionally, endoplasmic reticulum-resident selenoproteins also affect apoptotic responses in the presence of selenocompounds. Selenoproteins may also help cancer cells adapting against increased oxidative damage and the challenges of a modified nutrient metabolism that result from the Warburg switch. Finally, cancer cells may also rewire the selenoprotein hierarchy and use Se-related machinery to prioritize selenoproteins that are essential to the adaptations against ferroptosis and oxidative damage. In this review, we discuss both the evidence and the gaps in knowledge on how cancer cells from solid tumors use Se, Sec, selenoproteins, and the Se-related machinery to promote their survival particularly via resistance to ferroptosis.

## 1. Introduction: Selenium and Its Role in Solid Tumor Cancers

Selenium (Se) is an essential trace element that is required in an optimal concentration range by the human body to reduce health risks, including solid tumor cancer development. Above and below an adequate concentration range of Se, the risk of certain types of cancer, including skin [1], prostate [2], thyroid [3], breast [4], and lung [5] cancer, is increased. Although an association between Se status and cancer exists, it is unknown in detail about how Se is metabolized in cancer cells and solid tumors. Improved understanding of this process could generate more appropriate dietary recommendations that have thus far produced mixed results in the prevention of cancer, and enhance the effectiveness of targeted drug therapies.

Under physiological conditions, inorganic or organic Se is mostly metabolized into selenocysteine (Sec), an amino acid that is then inserted into the primary structure of peptide chains to form selenoproteins. The micronutrient Se is either metabolized through the trans-selenation pathway or via reduction by thioredoxin reductases (TXNRDs) in the presence of glutathione (GSH), depending on the dietary chemical form ingested [6]. Organic forms of Se include mostly selenomethionine (SeMet) or Sec while inorganic forms include mostly selenite or selenate. When Se levels are inadequate or deficient in cells, a postulated Se recycling mechanism may be triggered, coordinated by the enzyme Sec lyase (SCLY), which decomposes Sec into hydrogen selenide (H_2_Se) to be reutilized in selenoprotein synthesis [7].

The metabolism of Se may be manipulated by cancer cells, a consideration with implications to the precision nutrition concept used to support other cancer therapies [8]. Cancer cells can hijack Se metabolism in several ways. First, they may create solid tumor microenvironments that can have altered Se affinity even when the Se status of the patient is adequate [9,10]. Secondly, solid tumor cancers may also utilize traditional Se metabolic pathways such as the trans-selenation [11,12,13,14,15] or SCLY-mediated recycling to promote tumorigenesis [16,17], with yet few studies exploring this possibility. Finally, recent evidence suggests that cancer cells can hijack the ferroptosis pathway, a form of iron-mediated cell death, by increasing intake of Se, making them resistant to this form of cell death. Resistance to ferroptosis is achieved by turning the cancer cell into a selenophillic environment to promote the uptake of Se and Sec [9], which leads to upregulation of glutathione peroxidase 4 (GPX4), a key endogenous ferroptosis inhibitor [18]. The ability to resist to ferroptosis is a nascent, promising pathway target of future cancer therapeutics, that could reduce the survival of cancer cells.

Currently, the understanding of how cancer cells utilize Se beyond selenoprotein production is still limited. Particularly when considering that Se metabolism in cancer may be a target in precision nutrition to aid additional standard cancer therapies. In this review, we explore the gaps in current knowledge of the metabolism of Se and the role of selenoproteins and their synthesis machinery for the survival of cancer cells and cancer development. We focus on how Se status impacts solid tumors and how Se is metabolized in cancer. Finally, we explore how Se influences key selenoproteins in cancer, especially mechanisms related to GPX4-dependent ferroptosis.

### Selenium Deficiency and Toxicity in Cancer

In humans, the optimal range of Se in the serum is around 125 μg/L [19]. Excess Se can be toxic, causing either acute or chronic selenosis depending on the rate consumed. Acute selenosis occurs when an individual consumes a large dose of Se in a short period of time and is characterized by diarrhea, respiratory distress, vomiting, and potentially death [20,21]. Chronic selenosis occurs when the individual is exposed to a mildly high to high dose of Se over a long period, and the symptoms include fatigue, anemia, decrease in appetite, halitosis, and hair loss [20]. Se toxicity is also associated with an increased risk of certain cancers, such as skin [1] and prostate cancers [22]. Additionally, serum Se levels > 150 μg/L were associated with a modest increase in cancer mortality among adults in the United States. Paradoxically, the same study also noticed a rise in cancer mortality with serum Se levels below 130 μg/L, which could be considered optimal [23].

On the other side of the curve, inadequate amounts of Se or Se deficiency are also linked to an elevated risk of several diseases. These include Keshan’s Disease, a severe cardiomyopathy, autoimmune diseases, immune dysfunction, and certain cancers, such as thyroid [3], breast [4], and lung [5]. A 1985 study had already suggested a link between Se deficiency prior to diagnosis and cancer development [24]. Other studies since then have also noted the Se status of a patient influences cancer development. In a Poland cohort, Se levels below 60 μg/L were inversely associated with a risk of developing either lung or laryngeal cancer [5], an association that was also observed in a distinct cohort in which low Se status increased the risk of death after 10 years of diagnosis of breast cancer [4]. Altogether, optimal levels of Se around 125 μg/L prevent cancer development and mortality; however, it is still unclear how Se status affects localized Se metabolism within tumors and cancer cells.

## 2. Selenium Metabolism in Solid Tumor Cancers

### 2.1. Overview of Selenium Metabolism

In physiological conditions (Figure 1), dietary Se is mostly absorbed via the gastrointestinal tract and subsequently transported to the liver, where it is primarily metabolized [25]. In hepatocytes, the metabolism of Se varies depending on whether or not Se is in an inorganic or organic form. Inorganic forms such as selenite and selenate are reduced to H_2_Se by TXNRD enzymes in the presence of GSH to form selenodiglutathione (GSSeSG). Glutathione reductase converts GSSeSG to glutathioselenol (GSSeH), then GSSeH is then decomposed to H_2_Se by glutaredoxin [26]. Organic forms such as SeMet undergo a series of trans-selenation reactions. Cystathionine beta-synthase (CBS), cystathionine gamma-lyase (CGL), and SCLY participate in this trans-selenation pathway, producing Sec through SeMet metabolism [27]. CBS and CGL commonly produce sulfide (H_2_S) through the transsulfuration pathway [28], and intracellular Se metabolism uses the enzymes of the same system. CBS converts SeMet into selenocystathionine (SeCSE) and is subsequently converted into Sec by CGL. Finally, Sec is converted into H_2_Se by the enzyme SCLY to be delivered to selenophosphate synthethase 2 (SEPHS2) and translationally incorporated into selenoproteins as the amino acid Sec in its own tRNA, the Sec tRNA^[Ser]Sec^. Sec is the only known amino acid whose biosynthesis occurs on its own tRNA [29,30] and accompanied by the recoding of the UGA opal codon, generally defined as stop codon [31]. The production of selenoproteins is a prioritized, highly regulated mechanism, and excess Se is commonly either methylated or converted into a selenosugar to be excreted by the urinary system [32,33].

### 2.2. Selenium Metabolism in Solid Tumor Cancers

In cancer, the metabolism of Se is altered, although how tumorigenesis, invasiveness and malignancy impact this process is still unclear. Our current knowledge centers around the uptake of Se, particularly inorganic selenite, which is known to be cytotoxic to cancer cells and dependent on the x(c)− cystine transporter [34]. Intraperitoneal delivery of Se nanoparticles (SeNPs) containing selenite to cancer cells implanted into the peritoneal cavity of mice strongly killed these cells due to persistent generation of reactive oxygen species (ROS) [35]. This has the potential for cancer therapeutics as Se nanoparticles have already been shown to induce cancer cell apoptosis in four human cancer cell lines: A-172 (glioblastoma), Caco-2 (colorectal adenocarcinoma), DU-145 (prostate carcinoma), and MCF-7 (breast adenocarcinoma) [36]. Additionally, the organic Se-containing compound methylselenic acid (MSA) also demonstrated cytotoxic therapeutic potential via a mechanism that enhances ROS production and depletes GSH in cancer cells, producing methylselenol and bypassing SCLY [37]. Notably, Carlisle et al. demonstrated that several types of cancer cells are selenophilic, utilizing SLC7A11, a component of the cystine/glutamate antiporter SLC7A11, to promote the intake of Se and Sec [9]. The utilization of SLC7A11 to uptake Sec instead of cysteine facilitates the synthesis of selenoprotein GPX4 by cancer cells. GPX4 regulates ferroptosis, rendering protection against this cell death mechanism and enhancing survival. Se therapies that can deplete GSH such as MSA may bypass these protective mechanisms of cancer cells.

Remarkably, elevated intake of Sec results in the increased production of selenide, a toxic intermediate to cancer cells. To detoxify H_2_Se, cancer cells require selenoprotein SEPHS2. SEPHS2 is essential for the survival of cancer cells, as demonstrated by studies in which the loss of the *SEPHS2* gene in breast cancer cells (MDAMB231) impaired the growth of orthotopic mammary-tumor xenografts when these cells were injected in wild-type, nude athymic mice [9]. Potentially, this implicates SEPHS2 as a “weak spot” for a cancer cell, and therapeutics targeting this enzyme may blunt cancer cell growth and development. SEPHS2 is a key enzyme in selenoprotein synthesis, and its essential role in cancer development warrants further investigation, particularly to pinpoint specific molecular players involved in this essentiality, their regulation, and how they may affect the metabolism of Se in cancer.

Aside from the hijacking of the xCT transporter to promote intake of Se and Sec instead of cysteine in certain cancer cells, there are enzymes involved in the metabolism of Se, such as CGL, CBS, and SCLY, that may have roles in cancer cell survival and/or proliferation. These proteins and their roles in cancer will be discussed below.

### 2.3. Role of Cystathionine Gamma Lyase in Selenium Metabolism and Cancer

Cystathionine gamma lyase (CGL) is an enzyme that decomposes L-cysteine to produce H_2_S. The role of CGL has been well-documented in cancer. Several studies have linked CGL dysregulation with tumorigenesis in prostate [38], testicular [39], and breast [40,41] cancer, with promotion of their growth and development through enhanced CGL-mediated production of H_2_S [11,12]. A fusion protein with a mutated CGL to contain methionine gamma-lyase activity combined with annexin V was used to treat ovarian cancer cells growing in varying concentrations of SeMet [42]. The mutated CGL converted the SeMet into toxic methylselenol, generated ROS, and led to the death of these cells [42]. Paradoxically, this mechanism opposes the observations that H_2_Se promotes tumor growth [11,12] by increasing cell proliferation and preventing oxidative damage, and further clarification to understand this paradox is needed. Recent studies have focused on developing therapeutics to inhibit CGL and consequent excess H_2_S production. A novel CGL inhibitor named I194496 was capable of inhibiting the growth of human triple negative breast cancer cells by dual inhibition of the PI3K/Akt and Ras/ERK signaling pathways, and reduced metastasis via inhibition of the STAT3 and VEGF pathways [43]. Another inhibitor, I157172, reduced the growth, proliferation and migration of MCF7 breast cancer cells in a dose-dependent manner, accomplished by SIRT1-mediated deacetylation of STAT3 [44]. Nevertheless, these studies have focused on the involvement of CGL in H_2_S metabolism and its impact in cancer cells. Studies examining the effects of CGL inhibition on organic Se metabolism and consequent selenoprotein synthesis in cancer signaling pathways are still lacking. It is possible that the effects observed in the cancer signaling pathways involve selenoproteins and their control of redox status. Still, without a focus on evaluating Se effects, this possibility remains speculative.

### 2.4. Role of Cystathionine Beta-Synthase in Selenium Metabolism and Cancer

The first step in the transsulfuration pathway is conducted by the enzyme cystathionine beta-synthase (CBS), that enables the conversion of homocysteine (derived from methionine) to cystathionine and H_2_O and can also derive Sec from selenohomocysteine as part of the trans-selenation pathway [45,46]. As with CGL, most cancer studies have focused on the role of CBS in the production of H_2_S and how H_2_S supports tumor growth. There are several molecular mechanisms by which CBS-driven H_2_S production supports tumor growth, including via maintenance of mitochondrial respiration and ATP synthesis, stimulation of cell proliferation and survival, control of redox balance, and increasing vasodilation [47]. Elevated expression of CBS is found in several solid tumor-forming cells, including colon [48], ovarian [49], prostate [50], and breast cancer cells [51]. The role of CBS in cancer development has been well-reviewed elsewhere [47,52], and we will here focus on those studies that regard both Se and CBS.

Despite the well-documented link between suboptimal Se status and elevated risk for certain cancers, the physiological relevance of Se metabolic enzymes to cancer cell survival and proliferation remains unclear. As with CGL, the role of CBS in combination with Se is an understudied area. Among the few studies available, it has been shown that rats on Se-deficient diets present higher carcinogen-induced aberrant colon crypts and hypomethylated liver and colon DNA, possibly increasing tumorigenesis [13,14,15]. Notably, a study assessing the impact of Se deficiency on the hepatic metabolome of male mice found alterations in lipid and carbon-1 metabolism in the mouse liver due to a massive downregulation of CBS [27]. Strikingly, when animals were fed with half of the recommended levels of Se, total GPX and TXNRD activity was reduced and levels of taurine increased due to cysteine decomposition to taurine via the salvage pathway [27]. It is possible that changes to lipid and carbon metabolism contribute to the development of cancer in cases of suboptimal Se status; however, additional investigation is required to clarify the regulatory role of Se in this mechanism.

A study focusing on the effects of a Se-containing chrysin and folate targeted polyurea nanoparticle (Sechry@PUREG4-FA) on three ovarian cancer cell lines and two non-malignant ones [53] unveiled that SeChry had an inhibitory effect on CBS. This inhibitory effect prevented the production of H_2_S; however, the study has not assessed the specific effects of the Se moiety of SeChry on H_2_Se production. The compound also showed specificity towards the ovarian cancer cell lines and displayed reduced toxicity towards non-malignant cells. Although preliminary, this study brings a favorable light into the use of a Se nanoparticle in cancer therapy.

### 2.5. Role of Selenocysteine Lyase in Selenium Metabolism and Cancer

Unlike CGL and CBS, the role of SCLY in cancer is poorly understood. The enzyme SCLY generates H_2_Se by the decomposition of L-Sec into L-alanine [7], and the source of Sec is postulated to be either the trans-selenation or degradation of selenoproteins. A link between SCLY and mechanisms of cell proliferation, tumorigenesis or metastasis has been suggested previously [16]. A gene analysis study reported that the expression levels of SCLY were significantly increased in both human colon adenocarcinoma and esophageal carcinoma [16]. Despite uncovering an association of SCLY with these carcinomas, the sole use of bioinformatics approaches without wet lab confirmation to investigate several selenoproteins and their association with the top five most common cancers limits the conclusions for this study. Moreover, investigation of differentially expressed transcripts in livers of people with non-alcoholic fatty liver disease (NAFLD) found that SCLY was downregulated in non-alcoholic steatohepatitis (NASH) compared with healthy controls [17]. NAFLD and NASH are stages of hepatocellular carcinoma progression and are linked to obesity [54]. The downregulation of SCLY suggests a change in Se sources for selenoprotein synthesis occurs in NAFLD livers, further implicating SCLY as a potential target or localized biomarker for obesity-induced cancers, particularly in the liver [17]. Nevertheless, the specific role of SCLY in the tumorigenesis and progression of cancers derived from NAFLD or NASH, particularly at the early stages of cancer development, remains undetermined. SCLY may contribute to the Warburg metabolic switch in cancer cells by controlling selenoprotein synthesis, particularly the ones participating in energy metabolism and controlling carbohydrate or lipid utilization. Several selenoproteins are involved in energy metabolic pathways and cancer cell survival. SCLY, as a provider of H_2_Se for selenoprotein synthesis, may therefore contribute to tumor progression in tumors where selenoprotein synthesis is dampened.

SCLY has also been shown to connect Se and energy metabolism [55]. Whole body knockout (KO) of the *Scly* gene resulted in obesity, hepatic steatosis, hypercholesterolemia, hyperinsulinemia, and glucose intolerance with increased hepatic oxidative stress when Se levels were restricted [55]. Obesity is a well-known risk factor for cancer development [56,57,58], particularly esophageal adenocarcinoma, colorectal, liver, pancreatic, postmenopausal breast, endometrial, bladder and renal cell cancers [59,60,61,62,63]. Therefore, the participation of SCLY in gauging energy metabolism via the control of Se flux to selenoprotein synthesis turns this enzyme into a potential target in cancer research.

## 3. The Impact of Selenium on GPX4: Ferroptosis and Cancer

Increasing evidence suggests Se plays a significant role in ferroptosis, an iron-mediated cell death that is characterized by the accumulation of lipid peroxides that will eventually rupture the plasma membrane [64]. The main inhibitory pathway of ferroptosis described until now consists of the system Xc/GSH/GPX4 [65]. System X_c_− is the heteromeric antiporter that transports cystine into the cell. Cystine enters the cell to participate in the biosynthesis of GSH [66], which is a cofactor for the action of selenoprotein GPX4 [67]. GPX4 reduces toxic lipid peroxides into non-toxic lipid alcohols [65,68], preventing the accumulation of ROS. Hence, GPX4 is a primary regulator of ferroptosis as it directly curbs the accumulation of lipid peroxides [18]. GPX4 is a selenoprotein that requires Se to be synthesized, hence the ferroptosis response may be deemed Se-dependent [69]. Ferroptosis may be triggered when GPX4 is inactive, and ROS accumulates in the cell [69]. Remarkably, cancer cells increase the intake of Se and Sec to potentially control GPX4 levels and warrant themselves resistant to ferroptosis.

Recent studies have identified ferroptosis as a potential vulnerability for several types of cancer. Increased expression of LRP8, a low-density lipoprotein receptor that also functions as a receptor for selenoprotein P (SELENOP), was identified to promote ferroptosis resistance in breast cancer and hepatocellular carcinoma cell lines [10]. Once taken up by cells, SELENOP, which contains 10 Sec residues in humans and serves as a regulated Se source, is metabolized in lysosomes [70]. Following degradation of SELENOP, the Se-containing Sec is postulated to be captured by SCLY for its decomposition into H_2_Se, followed by delivery of H_2_Se to SEPHS2 for selenoprotein synthesis. Therefore, SELENOP provides cells with a pool of Se [71]. Elevated Se uptake via SELENOP suggests that cancer cells may require selenoprotein synthesis to render themselves resistant to ferroptosis. Corroborating this possibility, generated LRP8 KO breast and hepatocellular carcinoma cell lines presented reduced GPX4 levels, with concomitant reduction of Se levels that caused disruption in the GPX4 translation process. The disruption in translation that occurred with GPX4 was caused to ribosome stalling and early translation termination [10]. As GPX4 has traditionally been thought of as resistant to limited Se pools, this finding indicates that the selenoprotein hierarchy may be altered in cancer cells, resulting in an overreliance on GPX4. Thus, this overreliance is a potential weak spot of the proliferative mode in the cancer cell can be targeted through drug intervention to re-sensitize ferroptosis resistant cancers to become ferroptosis sensitive.

Expression levels of the system X_c_− antiporter determine the expression of GPX4 in breast cancer [72]. Strikingly, ferroptotic drug inducers erastin and RSL-3 also rendered breast cancer cells sensitive to ferroptosis and were utilized to explore how Se may be involved in the resistance to ferroptosis [72]. Treatment with the erastin depleted the levels of GPX4 and GPX1 by inhibiting xCT-dependent uptake of Se and rendered these cells vulnerable to lipid peroxidation and ferroptosis due to an overreliance to the xCT/GPX4 axis. Such vulnerability suggests that breast cancer cells may have an enhanced sensitivity to ferroptosis via several mechanisms. First, accumulation of H_2_Se formed during Sec decomposition is toxic to cancer cells as it increases ROS with these cells, therefore, relying on GPX4 to neutralize selenide-induced ROS. Second, cancer cells may be more susceptible to GPX4 inhibition due to a higher level of polyunsaturated fatty acid (PUFAs) and enhanced susceptibility to lipid peroxidation, which requires a steady need for GPX4. The final potential cause for ferroptosis sensitivity may be the increased antioxidant capacity of the cancer cells accompanied by a reduced requirement for pro-survival signaling activities, rendering these cells vulnerable when xCT/GPX4 protection is mitigated [72]. Over-reliance on GPX4 was again suggested when it was shown that the NCI-H295R adrenocortical carcinoma (ACC) cell line was also sensitive to RSL-3-induced ferroptosis, potentially due to the elevated expression of GPX4 [73]. Supporting this possibility, cells expressing a cysteine variant of GPX4 were shown to be highly sensitive towards peroxide-induced ferroptosis [74]. Mechanistically, these cells exhibited irreversible oxidation, as a critical cysteine residue, Cys46, in which GPX4 was oxidized to sulfonic acid (SO_3_H) upon H_2_O_2_ treatment, rendering the enzyme inactive. The change in oxidative state in the active site of GPX4 as a consequence of an amino acid change suggests that a steady Se supply is vitally necessary for the production of GPX4 in cancer cells, and their consequent resistance to peroxide-induced ferroptosis. It also reinforces that cancer cells may manipulate the system Xc/GSH/GPX4 to promote increased uptake of Se and control GPX4 levels. Figure 2 summarizes how cancer cells utilize Se to become resistant to ferroptosis.

Nevertheless, the connection between the Se supply and its utilization in cancer cells and their resistance to ferroptosis is complicated by the fact that selenite treatment induces cancer cells to undergo ferroptosis [34,75]. Immortal cancer cell lines found to be sensitive to selenite treatment via activation of ferroptosis, include derivatives of malignant glioma (U87MG), breast cancer (MCF-7), and prostate cancer (PC3). As cancers in general are highly heterogenous, it is very plausible to consider that certain cancer cells may be selenophobic while others may be selenophillic, resulting in certain solid tumors to be ferroptosis-sensitive while others may be ferroptosis-resistant. Further investigation of how Se allows for the distinct sensitivity profiles to ferroptosis in cancer cells is needed to potentially exploit these vulnerabilities into new therapies for at least a portion of solid tumors.

## 4. Impact of Selenoproteins on Cancer

Cancer cells are highly unique in that they have elevated metabolic rates due to their increased proliferation, creating higher levels of ROS and consequent oxidative stress [76]. The increased metabolic demand results in cancer cells being exposed to heterogeneous microenvironments, with some regions displaying a lack of key nutrients including glucose, and hypoxia [77]. To deal with the various microenvironments, selenoproteins involved in redox regulation, such as the GPXs and TXNRDs [68,78], are differentially regulated by cancer cells for their own benefit in comparison to healthy cells. This differential regulation of GPXs and TXNRDs possibly occurs to protect the cells from the oxidative stress generated by the increased metabolic demand. In this section, we explore how certain selenoproteins, particularly the GPXs and TXNRDs, impact specifically cancer cell biology.

### 4.1. Role of Glutathione Peroxidases (GPXs) in Cancer Metabolism

The GPX family of selenoproteins curbs oxidative stress in cells [68] and, in the case of GPX4, regulates ferroptosis by acting on lipid peroxidation of membrane systems [18]. Most of the GPXs have known roles in cancer and are reviewed elsewhere [79,80,81,82,83]. We focus here on how the Se metabolism influences cancer metabolism via GPXs aside GPX4, which has been discussed in the previous section.

In humans, five of 8 GPXs are selenoproteins and are dependent on the supply of Se. These five are GPX1, GPX2, GPX3, GPX4, and GPX6. The other three GPX homologs (GPX5, GPX7, and GPX8) have a cysteine in place of Sec [31]. GPXs catalyze the reduction of H_2_O_2_, organic hydroperoxides and lipid peroxides to yield lipid alcohols and water through the oxidation of two molecules of GSH [84]. This metabolic process is altered in cancer development [76] as many types of cancers including breast, ovarian, lung, head, and neck cancers have been reported to have elevated levels of GSH [85], and correspondingly, the GPXs [82]. Elevated GSH and GPXs may be a response to handle the increased oxidative stress that their high metabolic rates have added, and potentially connected to the heterogeneity of cancer responses [79,82]. Depending on the mutations present in the cancer cell, the heterogeneous tumor can either have cells downregulating or upregulating one of the GPXs, with cellular impacts on metabolic pathways of glucose [86], GSH [82], and lipid metabolism [87]. Table 1 summarizes which GPXs have been found to be up or downregulated in cancer.

#### 4.1.1. Role of Glutathione Peroxidases (GPXs) in Glucose Metabolism in Cancer

Cancer cells are known to alter their glucose metabolism to overcome enhanced ROS production [88]. High rates of aerobic glycolysis towards lactate production (Warburg effect) in human colon and breast cancer cell lines reduces ROS production by switching away from oxidative phosphorylation (OXPHOS) [89]. However, cancer cells will also enhance glucose flux through the pentose phosphate pathway (PPP) to generate more NADPH for antioxidant-related pathways in times of intense oxidative stress [90]. As GPXs detoxify ROS, it is therefore unsurprising that they are involved in detoxifying ROS generated by cancer cells. GPX1, a Se-sensitive selenoprotein whose complex roles in cancer have already been reviewed in detail [91], was reported to be involved in protective autophagy in pancreatic ductal adenocarcinoma (PDA) cells during glucose deprivation. GPX1 overexpression sensitized PDA cells to caspase-dependent apoptosis [86]. Loss of GPX2 in a metastatic breast cancer cell line resulted in increased malignancy due to upregulation of the HIF1*α* transcription factor and the stimulation of the Warburg effect, strengthening survival [92]. Furthermore, a single cell sequencing analysis showed the Warburg effect in metastatic breast cancer was found in all but one cell cluster, which exhibited the ability to use both glycolysis and OXPHOS. This exception further demonstrates the complex heterogeneity when attempting to understand the biology of cancer [86,89]. It also suggests that, when the Warburg effect occurs, GPXs may be downregulated due to less ROS production, and when OXPHOS is still prevailing, GPXs may be upregulated to curb the excess ROS [89].

#### 4.1.2. Role of Glutathione Peroxidases (GPXs) in Glutathione and Lipid Metabolism in Cancer

Glutathione (GSH) and lipid metabolism are closely linked and, therefore, will be discussed together. It has been observed in several cancers, including breast, ovarian, lung, head, and neck, that GSH levels are elevated [85]. As previously discussed in this review, one of the most essential pathways in which GPX is involved in the detoxification of lipid peroxides through GPX4. Inhibition of GPX4 leads to depletion of GSH, resulting in lipid peroxidation and ferroptosis [18]. However, GPX4 is not the only regulator of GSH, as other GPXs catalyze the reduction of H_2_O_2_ and lipid hydroperoxides while expending GSH as well [84]. GPX3, which can be found in the extracellular space and in circulation, is downregulated in several cancers including bladder, breast, cervical, lung, and prostate cancers (summarized in Table 1), The downregulation of GPX3 is negatively associated with increased lipid peroxidation [80]. Interestingly, in patients with metastatic cancer, total GPX activity, Se, and zinc levels were reported to be significantly lower than in age-matched controls [93], suggesting that GPX3 may function as a tumor suppressor, protecting against systemic oxidative stress. As such, numerous studies have focused on the role of GPX3 in cancers [80].

The involvement of other GPXs in lipid metabolism in cancer is less known. Nevertheless, combined deletion of Gpx1 and Gpx2 in mice resulted in the development of colon cancer, deemed occurring due to the inability of cells to reduce toxic peroxides at the expense of GSH [94]. GPX6 is particularly understudied, and if this enzyme has a role in cancer, it remains to be revealed.

### 4.2. Thioredoxin Reductases (TXNRDs) and Cancer

The TXNRD family of enzymes are known for maintaining redox homeostasis utilizing thioredoxin as substrate [95]. TXNRDs and their role in cancer have also been reviewed elsewhere [96,97,98]. This section will be focused on TXNRD1 as it has been the main TXNRD studied in cancer, and we will briefly comment on TXNRD2 and TXNRD3 and their relationship to cancer metabolism.

#### 4.2.1. TXNRD1 and Oxidative Stress in Cancer

TXNRD1 is a critical selenoprotein in cancer cells [95], and found to be overexpressed in several cancer types when compared to TXNRD2 and TXNRD3 [99]. Table 1 depicts the regulation of the TXNRD1 gene according to the type of cancer. Previous studies have suggested TXNRD1 as a prognostic tool for cancers such as hepatocellular carcinoma (HCC), lung, and breast cancer [99]. Higher expression of this enzyme in these cancers when compared to healthy cells was associated with a poor prognosis for patients, suggesting that TXNRD1 is an oncogene. Hence, cancer therapies focusing on inhibiting TXNRD1 to increase oxidative stress and promote cell death are a current focus of research [100]. Indeed, several studies using TXNRD1 inhibitors have supported this view [101,102,103]. For instance, the pharmacological pan-TXNRD inhibitor auranofin (AUR) prevented the growth of HCC tumors in mice [101]. The natural product santamarine also achieved inhibition of TXNRD1 in HeLa cells, a cervical cancer cell line, causing these cells to undergo oxidative stress-mediated apoptosis [102]. Finally, the drug cynaropicrin, a natural product found in leaves of artichokes, also induced apoptosis of HeLa cells through inhibition of TXNRD1 [103]. Therefore, similarly to the GPXs, it is likely that the high metabolic demand and increased oxidative stress results in cancer cells upregulating TXNRD1 to combat elevated ROS, deeming TXNRD1 an appropriate target for therapeutic development.

Aside from direct inhibition of TXNRD1, other studies in cancer have focused on the regulation of TXNRD1 through its transcription factor nuclear factor-erythroid factor 2 (Nrf2). Nrf2 positively regulates TXNRD1 expression [104]. In HCC, it was found that there is a reduced amount of the gluconeogenesis enzyme, phospholenolpyruvate carboxykinase (PEPCK1) which causes elevated levels of cellular ROS and results in the activation of Nrf2 and upregulation of TXNRD1 [105]. Furthermore, non-small cell lung cancer (NSCLC) tumors with nuclear accumulation of Nrf2 were associated with increased TXNRD1, chemoresistance, and poor prognosis [106]. Therefore, Nrf2 control of TXNRD1 expression in cancer may also be considered when targeting TXNRD1 to promote oxidative-stress induced cell death in cancer cells.

#### 4.2.2. TXNRD1 and Ferroptosis in Cancer

Recent evidence suggests that TXNRD1 may also be involved in ferroptosis resistance in cancer cells. Nevertheless, the focus of the field remains on GPX4, with the role of TXNRD1 in ferroptosis being understudies. Still, several lines of evidence suggest that TXNRD1 is involved in regulating ferroptosis in cancer [98]. First, it is known that GSH deficiency can be rescued by combined expression of the cysteine-glutamate antiporter and TXNRD1 [66]. High doses (25 mg/kg) of the drug AUR, inhibited total TXNRDs and induced lipid peroxidation and ferroptosis in a mouse model of hemochromatosis. Second, a TXNRD1 specific inhibitor, TRi-1 also resulted in an increase in hepatic lipid peroxidation that was completely blocked with the ferroptosis inhibitor ferrostatin-1 [107]. Third, the alkaloid piperlongumine, extracted from long peppers and known TXNRD1 inhibitor, was able to sensitize human breast cancer cells (MCF-7) and human lung cancer cells (A549) to ferroptosis using the known ferroptosis inducer, erastin [108]. Erastin inhibits the cysteine-glutamate antiporter, thereby depleting the cells of GSH and resulting in downstream lipid peroxidation and ferroptosis. Hence, both TXNRD1 and GPX4 are capable of regulating ferroptosis. Nevertheless, as the involvement of TXNRD1 in ferroptosis is a novel finding, there are still details to explore and uncover, particularly in relation to its role in cancer cell growth and development.

#### 4.2.3. Role of TXNRD2 and TXNRD3 in Cancer

It is currently unclear the involvement of TXNRD2 and TXNRD3 in cancer. Nevertheless, most studies have found both enzymes to have positive correlations with tumors thereby suggesting that these TXNRDs may act as oncogenes, like TXNRD1 [100,109,110]. The specific physiological role of TXNRD2 and TXNRD3 in cancer cells remains to be determined, with some recent studies starting to enlighten their role separately [111]. Deficiency of TXNRD2, in immortalized embryonic fibroblasts from mice resulted in prolyl hydroxylase 2 (PHD2) stabilization and inhibited HIF-1*α* and vascular endothelial growth factor (VEGF) signaling, leading to impaired tumor growth and angiogenesis [112]. Using an shRNA approach, there was also reduction of proliferation and metabolism in, NSCLC cell lines (A549 and NCI-H1299) in which when TXNRD2 was downregulated, of the cells [113]. Furthermore, loss of TXNRD2 in the same cells also increased the production of ROS and led to apoptosis.

The role of TXNRD3 in cancer development is less known than the role of TXNRD2. It has been shown that knockout of TXNRD3 in mice resulted in increased severe ulcerative colitis and lesions than in corresponding wild-type mice, conditions that are known to be a risk factor for the development of colorectal cancer [114]. Furthermore, overexpression of TXNRD3 in a murine colon cancer cell line (CT26) increased oxidative stress and necrosis in these cells [114]. Despite limited evidence, these findings intimate that TXNRD2 and TXNRD3 also inhibit oxidative stress via the thioredoxin system in cancers as TXNRD1, thereby promoting cancer cell resistance to cell death.

### 4.3. Endoplasmic Reticulum-Residents Selenoproteins in Solid Tumor Cancers

Persistent endoplasmic reticulum (ER) stress has been shown to promote tumor growth and metastasis [115]. There are seven selenoproteins localized to the ER [116]: selenoproteins F (SELENOF), K (SELENOK), M (SELENOM), N (SELENON), S (SELENOS), T (SELENOT), and type 2 iodothyronine deiodinase (DIO2). Most of these selenoproteins are involved in misfolded protein degradation, calcium homeostasis mechanisms or thyroid hormone activation, but with an unclear role in cancer cell protection against cell death.

Recently, an evaluation of the expression of these seven selenoproteins in human cancer cell lines DU145 (prostate carcinoma), MCF7 (breast adenocarcinoma) and HT-1080 (fibrosarcoma) after treatment with the Se-derived anti-cancer agent MSA [117,118] revealed that the expression pattern of SELENOM is opposite to SELENOT and SELENOF, which implicates the prioritization of Se and selenoprotein synthesis to allow for sorting according to the cell state [119]. Moreover, the same study showed that DIO2, SELENON, SELENOK and SELENOS were upregulated upon treatment with the highest concentration of MSA. Notably, MSA treatment caused activation of apoptotic pathways in all three cancer cell lines, findings that highlight the complex and dual potential role of Se compounds in future therapeutics, that warrants further understanding, particularly on how to prioritize the production of specific selenoproteins.

Knockdown of SELENOM and SELENOT, two selenoproteins that play a critical role in brain development and have increased expression in brain cancers [120,121], leads to ER stress in A-172 cells, a human glioblastoma cell line [116]. Upon ER stress activation, SELENOM knockdown resulted in increased levels of several pro-apoptotic proteins including CHOP, GADD34, PUMA, and BIM, while knockdown of SELENOT had the opposite effect and reduced the expression of these same pro-apoptotic proteins [116]. However, glioblastoma cells in which SELENOT was knocked down and treated with MSA combined with Se nanoparticles (SeNPs) resulted in apoptotic cell death [116]. It is likely that both SELENOT and SELENOM protect cancer cells from ER stress-induced cell death, but further clarification as well as the translatability of these findings to cancer prevention or therapeutics still need to be uncovered.

### 4.4. Synthesis of Selenoproteins in Cancer Cells

Besides the GPX and TXNRD selenoprotein families, other selenoproteins have also been shown to be implicated in various stages of cancer development. Notably, several studies have assumed their participation in cancer cell survival and proliferation, tumorigenesis, invasiveness, and metastasis processes based on transcriptomics analysis, which raises significant limitations to their impact in prognosis or therapy and may serve as a initial step towards functional determination of the role of each of these proteins in cancer development. Table 1 discloses a non-comprehensive, concise overview of transcriptomics findings for human selenoprotein genes in various cancer types, and their prognostic value. The prognostic value was obtained from The Human Protein Atlas database, which analyzed a large transcriptomic database of tumor samples of 17 cancer types from Sweden [122]. The prognostic value is displayed here to highlight the great divergence related to the potential use of certain selenoproteins as prognostic biomarkers in cancer.

**Table 1 biomolecules-12-01581-t001:** Selenoproteins and their regulation (↑ upregulation, ↓ downregulation) and prognostic value in cancer [123,124].

Selenoprotein	Regulation/Cancer	Prognostic Value (Favorable/Non-Favorable)	References
DIO1	↓ colon adenocarcinoma, hepatocellular carcinoma, lung squamous cell carcinoma, papillary thyroid carcinoma, anaplastic thyroid carcinoma, prostate cancer, oesophageal carcinoma; ↑ mammary gland carcinoma	NF: thyroid cancer	[16,125,126,127,128]
DIO2	↑ colon adenocarcinoma, hepatocellular carcinoma, lung adenocarcinoma and lung squamous cell carcinoma, pituitary adenoma, oligodendroglioma, gliosarcoma, glioblastoma multiforme; ↓ astrocytoma, glioblastoma	F: endometrial, colorectal cancer	[16,129,130]
DIO3	↓ lung adenocarcinoma, astrocytoma; ↑ hepatic hemangioendothelioma, gliosarcoma, glioblastoma multiforme, adenoma and colon carcinoma	ND	[16,129,131,132,133]
GPX1	↓ lung squamous cell carcinoma	F: endometrial cancer	[16]
GPX2	↑ colon adenocarcinoma, lung adenocarcinoma and lung squamous cell carcinoma, non-triple negative breast cancer	NF: head and neck cancer	[16,134]
GPX3	↓ colon adenocarcinoma, oesophageal carcinoma, lung adenocarcinoma and lung squamous cell carcinoma, stomach adenocarcinoma, thyroid carcinoma, non-triple negative breast cancer	NF: stomach cancer	[16,128,134]
GPX4	↑ thyroid, renal, prostate, cervical, colon adenocarcinoma, endometrial, melanoma, lung, non-triple negative breast cancer	F: endometrial, cervical, pancreatic, renal cancers, lung, NF: colon adenocarcinoma	[16,134,135]
GPX6	↑ non-triple negative breast cancer	ND	[134]
MSRB1	↑ colorectal cancer, hepatocellular carcinoma	ND	[136,137]
Selenoprotein F	↑ colorectal cancer	F: Colorectal cancer; NF: liver cancer, head and neck cancer	[138]
Selenoprotein H	↑ colorectal cancer	F: ovarian cancer; NF: liver cancer	[139]
Selenoprotein I	↑ colon adenocarcinoma, lung adenocarcinoma, lung squamous cell carcinoma, melanoma	ND	[16,140]
Selenoprotein K	↑ hepatocellular carcinoma	F: lung cancer, endometrial cancer	[141]
Selenoprotein M	↑ hepatocellular carcinoma; ↓ stomach adenocarcinoma	NF: renal cancer	[16]
Selenoprotein N	↑ hepatocellular carcinoma; ↓ melanoma	F: endometrial cancer; NF: liver cancer	[16,140]
Selenoprotein O	↑ lung squamous cell carcinoma; ↓ thyroid cancer	F: Urothelial cancer	[16,128]
Selenoprotein P	↓ hepatocellular carcinoma, lung adenocarcinoma, lung squamous cell carcinoma, thyroid cancer, colitis-associated cancer	F: Renal cancer	[16,140]
Selenoprotein S	↓ thyroid cancer; ↑ non-triple negative breast cancer	F: endometrial, ovarian cancer	[128,134]
Selenoprotein T	↑ melanoma	ND	[140]
Selenoprotein V	↑ lung squamous cell carcinoma; ↓ thyroid cancer	NF: thyroid cancer	[16,128]
Selenoprotein W	↓colon adenocarcinoma, stomach adenocarcinoma, melanoma	F: lung cancer	[16,140]
SEPHS2	↑ lung adenocarcinoma, thyroid, colorectal cancer, adrenocortical carcinoma, bladder urothelial carcinoma, breast invasive carcinoma, glioblastoma multiforme, lower grade glioma; ↓ breast cancer, renal papillary cell carcinoma	F: renal cancer; NF: glioma	[16,142,143]
TXNRD1	↑ hepatocellular carcinoma, lung adenocarcinoma, lung squamous cell carcinoma, melanoma, non-triple negative breast cancer	NF: liver, hepatocellular carcinoma, and renal cancer	[16,99,134,140]
TXNRD2	↑ colorectal cancer, hepatocellular carcinoma, non-small cell lung cancer, renal cancer	F: renal cancer	[16,113,144]
TXNRD3	↑ non-triple negative breast cancer, colorectal cancer	F: renal cancer; NF: pancreatic, endometrial cancer	[110,134]

The role of each selenoprotein in cancer is commonly determined by how its molecular function contribute to cell survival in a proliferative environment. Most selenoproteins have had their molecular function or regulatory role in cancer cell-related processes discussed elsewhere [123] with a few of them remaining understudied. Notably, the molecular mechanism of SELENOF and SELENOP in colorectal cancer development have been detailed using transgenic mouse models that lacked these enzymes [138,145]. Nevertheless, even accounting the many unknowns, the broad diversity of roles that this group of proteins display and the nature of their role in cancer cell according to developmental stage, genotypical variation, and prognostic and diagnostic value suggest a highly regulated fine tuning for selenoprotein synthesis in general. This fine tuning may be predominantly dependent to the microenvironment that prevails according to each cancer type and especially the dependence on Se for cell proliferation and metabolic survival. It could also reflect fine tuning of Se metabolism affecting the selenoprotein hierarchy.

To that matter, a recent study evaluated the co-essentiality network of the selenoprotein synthesis machinery within 485 human cancer cell lines and revealed a tight cluster encompassing genes for the selenoprotein synthesis machinery and three selenoproteins, SEPHS2, GPX4 and TXNRD1, suggesting that the machinery functions in most cancer cells to support the production of these selenoproteins [95]. Se taken up by human cancer cells, therefore, should be mostly used to produce these three selenoproteins. Available Se is prioritized to the synthesis of certain selenoproteins according to tissues and metabolic demands of a cell, and particularly under Se deficiency, the hierarchy would be expected to prevail [146,147]. Nevertheless, mechanisms of rewiring of the hierarchy have been described in cancer cells, dependent on SECIS element recoding efficiency and ribosome stalling during translation of GPX4 when Se levels are low, reducing GPX4 levels [10,148]. This occurrence suggests nuances of the molecular mechanisms of selenoprotein synthesis in cancer cells. Furthermore, upstream Se metabolism, i.e., the uptake and break down of the different Se chemical forms in cancer cells prior to delivery to the selenoprotein synthesis machinery, also have molecular mechanisms of recognition that will allow for prioritization of synthesis of selenoproteins most needed for cancer cell survival and proliferation, as discussed previously in this review. The strong dependence of cancer cells on Se activates detoxification mechanisms of H_2_Se, the end product of upstream Se metabolism, that are carried out by SEPHS2, one of the prioritized selenoproteins in these cells [9]. Additional recognition/prioritization mechanisms may occur in cancer cells, contributing to a steady availability of Se for the survival and proliferation.

## 5. Conclusions

The requirement of Se in its various chemical forms by cancer cells and its consequent metabolism may become a potential therapeutical target. The metabolism of Se, Sec and the synthesis/degradation of selenoproteins participate in promoting resistance to cell death and warranting metabolic homeostasis in cancer. Cancer cells may increase the uptake of Se through system X_c_− or as SELENOP through LRP8 receptor to protect the cell from ferroptotic cell death, while also modulating key enzymes of the trans-selenation pathway, such as CBS, CGL, or SCLY. The primary biological use of Se in cells is to produce selenoproteins, which are involved in modifying glucose and lipid metabolism to protect the cancer cell from oxidative damage. The recent finding of the prioritization GPX4, TXNRD1, and SEPHS2 synthesis in cancer cells, favoring a cell proliferative phenotype, repurposes the selenoprotein synthesis hierarchy in cancer cells. These particular selenoproteins are likely prioritized due to their importance in preventing ferroptotic cell death, curbing oxidative stress, and allowing for selenoprotein synthesis. Nevertheless, selenoproteins can both be detrimental or protective to the cancer cell, depending on what mutations are accumulated and the type of cancer, and this distinct profile should be manipulated accordingly in future therapeutics. Recent evidence has suggested potential vulnerabilities involving Se metabolism in cancer cells that may also grant functional targets for the development of novel therapies, particularly in the case of Se nanoparticles and Se-containing compounds such as methylselenic acid, with known cytotoxic effects on cancer cells. These Se nanoparticles and selenocompounds may be used in combination with immunotherapies or even standard of care therapies, to ameliorate the Se status of a cancer cell. Understanding of how cancers utilize Se to their survival advantage may lead to the development of novel Se-based therapeutics or precision nutrition recommendations that target both Se-dependent mechanisms and additional metabolic pathways that may be regulated by Se.

## 6. Methods

The following criteria were used to select references for this review:

Databases searched: Pubmed (NLM)

Keywords searched: Selenium, cancer, selenocysteine lyase, selenium deficiency, selenium toxicity, cystathionine gamma lyase, cystathionine beta synthase, solid tumors, ferroptosis, glutathione peroxidase, thioredoxin reductase, selenoproteins

## Figures and Tables

**Figure 1 biomolecules-12-01581-f001:**
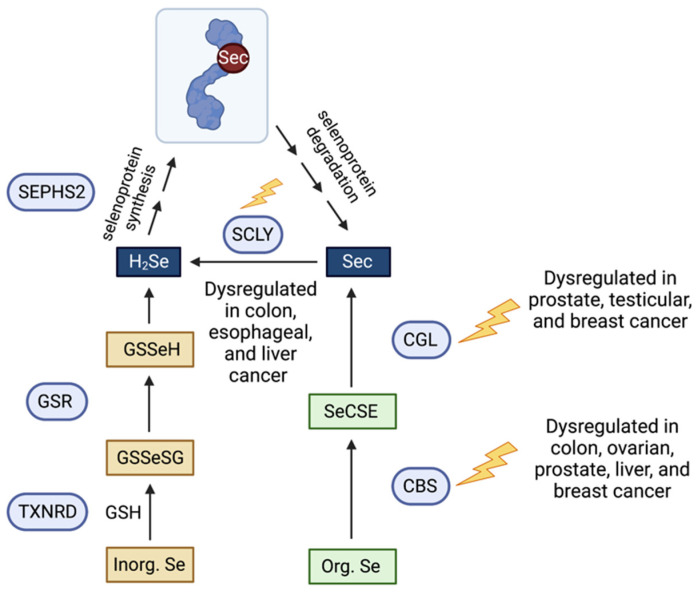
Schematic representation of the organic and inorganic Se metabolic pathways, Sec decomposition and their dysregulation in cancer. Arrows indicate direction of reaction. Blue circles represent enzymes; squares represent selenometabolites. GSH, glutathione; GSR, gluthathione reductase; TXNRD, thioredoxin reductase; GSSeSG, selenodiglutathione; GSSeH, glutathioselenol; H_2_Se, hydrogen selenide; SEPHS2, selenophosphate synthetase 2; Sec, selenocysteine; SCLY, selenocysteine lyase; CGL, cystathionine gamma lyase; CBS, cystathionine beta synthase; SeCSE, selenocystathionine.

**Figure 2 biomolecules-12-01581-f002:**
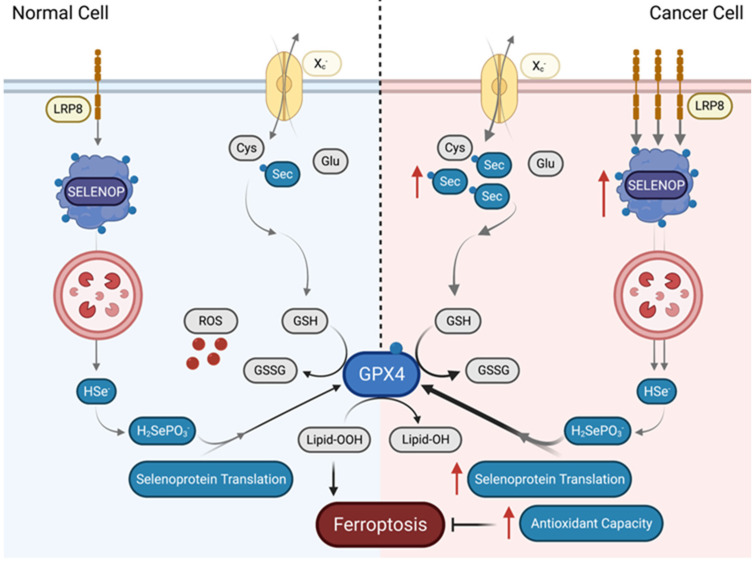
Cancer cells resist ferroptosis by increasing uptake of Se. In normal cells, ferroptosis is triggered after accumulation of toxic lipid peroxides. Certain cancer cells, however, can render themselves resistant to ferroptosis through increased uptake of Se either through the X_c_− antiporter in the form of Sec or through increased uptake of SELENOP through LRP8 into lysosomes. In the lysosome, Se is then postulated to be released via SCLY decomposition of SELENOP in order to provide the cancer cell with a pool of Se. This increases GPX4 expression through increased selenoprotein translation, and thereby expands the antioxidant capacity of the cancer cells, rendering them resistant to ferroptosis. Sec, selenocysteine; X_c_—, cysteine-glutamate antiporter; Glu, glutamate; LRP8, LDL receptor related protein 8; SELENOP, selenoprotein P; GPX4, glutathione peroxidase; Cys, cysteine; HSe–, hydrogen selenide; GSH, glutathione, GSSG, glutathione disulfide; H_2_SePO_3_, selenophosphate; Lipid-OOH, lipid peroxide; Lipid-OH, lipid alcohol.

## Data Availability

Not applicable.

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
