# Peer review of "Metabolism of Selenium, Selenocysteine, and Selenoproteins in Ferroptosis in Solid Tumor Cancers"

_biomolecules, 2022, doi:10.3390/biom12111581_

Round 1

Reviewer 1 Report

This review is an interesting topic that comprehensively collected information about selenium metabolism and selenoproteins. The authors tried to focus on the role of selenium in ferroptosis and redox homeostasis. However, this manuscript is not novel and the topic is too broad. I would suggest the authors narrow down the topic and their manuscript. Overall, I recommend the manuscript be published after a major correction.

Comments

Title: authors used a very broad title, but they have not provided information about redox homeostasis. I suggest narrowing down the title to present what authors have collected in the context.

Abstract, authors should give more explanation of what they collected in the review. Abstract should be a summary of information in the article.

The style and English used in this article are more suitable for book chapter writing, not for scientific article publishing. For example, there are lots of redundant words and sentences that have been used. Authors used a number of vague pronouns "These, that etc" which is hard for readers to find where they referred to. Here are some examples

Page 2, line 80: in Poland cohort..

page 2, line 83-84: following diagnosis within a different polish cohort..

page 2, line 86-7: " in the next section, we will discuss..." this is a redundant sentence since the readers will automatically follow to the next section so there is no need to explain!

page 3, line 137:  "piggyback"..not a suitable word in scientific writing.

page3, line 138: redundant sentence.

page 4, line 159: the word "used" is typed twice.

page 8, lines 323-324: redundant sentence.

page 9, line 392-393: redundant sentence.

words like "encouragingly" and "interestingly" are emotional and not commonly used in scientific writing. I would suggest the authors replace these types of words with fact-based information.

There are lots of sentences and paragraphs throughout the manuscript which have been left without referencing. It is unclear if these sentences are the authors' speculations or conclusions. for example

page 4, lines 165-170: reference is required

page 5, lines 218-227: reference is required

page 8, lines 341-342: reference is required

page 8, lines 360-365: reference is required.

page 10, lines 440-445: reference is required.

page 10, line 460: reference is required.

Table 1: it is a good effort to collect all the data in a table, but the authors need to explain what "arrows" mean in the title! I also suggest switching the place of the column of references with the column of "prognostic value".

Conclusion:

page 13, lines 528-531: this is not a conclusion of your article, authors need to summarize what they explained in the article and the prospect of selenium metabolism studies or how they can be used in cancer therapy.

Author Response

Reviewer #1

This review is an interesting topic that comprehensively collected information about selenium metabolism and selenoproteins. The authors tried to focus on the role of selenium in ferroptosis and redox homeostasis. However, this manuscript is not novel and the topic is too broad. I would suggest the authors narrow down the topic and their manuscript. Overall, I recommend the manuscript be published after a major correction.

Response: We thank the reviewer for finding the topic of the review interesting and appreciate the reviewer for their insightful and helpful suggestions to improve the manuscript. We have addressed the reviewer’s comments as detailed below.

Specific Comments:

  1. Title: authors used a very broad title, but they have not provided information about redox homeostasis. I suggest narrowing down the title to present what authors have collected in the context.

Response: We thank the reviewer for their helpful suggestion and have changed the title to reflect more of what was collected in the review (page 1).

  1. Abstract, authors should give more explanation of what they collected in the review. Abstract should be a summary of information in the article.

Response: We agree with the reviewer that the abstract should reflect more of what was collected in the review. We have now revised the abstract accordingly, focusing more on the topics discussed in the review (page 1, L. 8-25).

  1. The style and English used in this article are more suitable for book chapter writing, not for scientific article publishing. For example, there are lots of redundant words and sentences that have been used. Authors used a number of vague pronouns "These, that etc" which is hard for readers to find where they referred to. Here are some examples

Page 2, line 80: in Poland cohort..

page 2, line 83-84: following diagnosis within a different polish cohort..

page 2, line 86-7: " in the next section, we will discuss..." this is a redundant sentence since the readers will automatically follow to the next section so there is no need to explain!

page 3, line 137:  "piggyback"..not a suitable word in scientific writing.

page3, line 138: redundant sentence.

page 4, line 159: the word "used" is typed twice.

page 8, lines 323-324: redundant sentence.

page 9, line 392-393: redundant sentence.

words like "encouragingly" and "interestingly" are emotional and not commonly used in scientific writing. I would suggest the authors replace these types of words with fact-based information.

Response: We commend the reviewer for his/her helpful suggestions to improve the style and English of the manuscript and have now rewrote the manuscript. We have either removed emotionally-attached adverbs, or substantitally edited the words and redundancies as suggested by the reviewer. We also used a commercially available English software to gauge the appropriateness of syntax and grammar changes. Please see the revised language throughout the manuscript.   

  1. There are lots of sentences and paragraphs throughout the manuscript which have been left without referencing. It is unclear if these sentences are the authors' speculations or conclusions. for example

page 4, lines 165-170: reference is required

page 5, lines 218-227: reference is required

page 8, lines 341-342: reference is required

page 8, lines 360-365: reference is required.

page 10, lines 440-445: reference is required.

page 10, line 460: reference is required.

Response: We thank the reviewer for their thorough reading of our review and apologize for the oversight of not including references where they are required. We have now added references as indicated throughout the manuscript. In the uploaded marked version, the newly-added references are highlighted in yellow.

  1. Table 1: it is a good effort to collect all the data in a table, but the authors need to explain what "arrows" mean in the title! I also suggest switching the place of the column of references with the column of "prognostic value".

Response: We appreciate the reviewer’s suggestion to explain what the arrows mean in the title and have now added this to the table title. We also agree with the reviewer regarding switching the reference column and the prognostic value column and have now modified the table accordingly. Please see pages 12-14 Table 1.

Conclusion:

page 13, lines 528-531: this is not a conclusion of your article, authors need to summarize what they explained in the article and the prospect of selenium metabolism studies or how they can be used in cancer therapy.

Response: We appreciate the reviewer for their recommendation to improve the conclusion of the article and have revised the conclusion accordingly. Please see now page 15 L. 588-612.

Reviewer 2 Report

The authors present a very thorough and detailed review of selenoproteins and their impact on ferroptosis, metabolism as it relates to cancer. The review is well-written and provides appropriate context and background. I have a few minor suggestions:

1) I would suggest that the authors include a schematic/figure to complement the section on Selenium metabolism. This is a key component and would be useful to the uninformed reader of this review.

2) I wonder if terminology should be updated in the figure to refer to SEPP1 as SELENOP? Not a strong feeling on this.

3) The left half of Figure 1 could be better described in the text, including the considerartion for other SELENOP receptors, affect on other SEPs, and ROS. I think these components can be added to the figure, while still emphasizing the main point related to GPX4 and ferroptosis. 

4) Table 1 is excellent. One small typo: collitis--> should be colitis

Author Response

Reviewer #2

The authors present a very thorough and detailed review of selenoproteins and their impact on ferroptosis, metabolism as it relates to cancer. The review is well-written and provides appropriate context and background. I have a few minor suggestions:

Response: We thank the reviewer for their kind comments regarding our review and finding it very thorough, well-written, and detailed. We have addressed the reviewer’s comments as suggested below.

  • I would suggest that the authors include a schematic/figure to complement the section on Selenium metabolism. This is a key component and would be useful to the uninformed reader of this review.

Response: We appreciate the reviewer for this helpful suggestion and have now added an additional figure to complement the section on selenium metabolism. Please see page 3, Figure 1.  

2) I wonder if terminology should be updated in the figure to refer to SEPP1 as SELENOP? Not a strong feeling on this.

Response: We thank the reviewer for this suggestion and have now revised the figure to be SELENOP to make the terminology consistent with the text and according to Gladyshev et al, JBC 2015, which defined the nomenclature for selenoprotein genes in humans. Please see page 8, figure 2.

3) The left half of Figure 1 could be better described in the text, including the considerartion for other SELENOP receptors, affect on other SEPs, and ROS. I think these components can be added to the figure, while still emphasizing the main point related to GPX4 and ferroptosis

Response: We commend the reviewer for this helpful suggestion and have revised the text to explain the left half of the figure more clearly (page 7-8) as well as modified the figure (page 8).

4) Table 1 is excellent. One small typo: collitis--> should be colitis

Response: We apologize for this typo and thank the reviewer for finding it. We have now corrected the table to be colitis (page 13).

Reviewer 3 Report

1) The abstract needs to be rewritten. In its present form, it does not reflect the essence of the work. The abstract should be written in more detail.

2) Endoplasmic reticulum resident selenoproteins should be discussed in detail. Especially SELENOM and SELENOT. https://pubmed.ncbi.nlm.nih.gov/35741332/

3) The role of selenium-containing compounds and selenium nanoparticles in cancer treatment should be discussed. https://pubmed.ncbi.nlm.nih.gov/34205571/, https://pubmed.ncbi.nlm.nih.gov/34360564/

4) One figure is not enough as an illustration for a review.

Author Response

Reviewer #3

We thank the Reviewer for their useful comments and suggestions. We have addressed the Reviewer’s comments as detailed below.

  • The abstract needs to be rewritten.In its present form, it does not reflect the essence of the work. The abstract should be written in more detail.

Response: We agree with the Reviewer and note that this point was also raised by Reviewer #1. We agree that the abstract should reflect more of what was collected in the review. We have now revised the abstract accordingly, focusing more on the topics discussed in the review (page 1, L. 8-25).

  • Endoplasmic reticulum resident selenoproteins should be discussed in detail.Especially SELENOM and SELENOT. https://pubmed.ncbi.nlm.nih.gov/35741332/

Response: We thank the Reviewer for this suggestion and have added a section on ER resident selenoproteins, with a paragraph dedicated to SELENOM and SELENOT. Please refer to page 12, L. 506-535.

  • The role of selenium-containing compounds and selenium nanoparticles in cancer treatment should be discussed. https://pubmed.ncbi.nlm.nih.gov/34205571/, https://pubmed.ncbi.nlm.nih.gov/34360564/

Response: We agree with the Reviewer and have now included a discussion of selenium containing compounds and selenium nanoparticles in cancer treatment (page 4, L. 134-149), as well a description of various studies in vitro using methylselenic acid and assessing for ER-resident selenoproteins (p. 12, L. 507-529).

4) One figure is not enough as an illustration for a review.

Response: An additional figure was also suggested by Reviewer #2, and we accomodated a 2nd figure in the review as suggested by both Reviewers. Thank you for the suggestion.

Round 2

Reviewer 1 Report

The authors have improved the manuscript based on the comments. Therefore, I would suggest the acceptance of their manuscript for publication in the Biomolecules.

Reviewer 3 Report

The article has been significantly improved and can be accepted for publication.